# PI3K/mTOR Dual Inhibitor Pictilisib Stably Binds to Site I of Human Serum Albumin as Observed by Computer Simulation, Multispectroscopic, and Microscopic Studies

**DOI:** 10.3390/molecules27165071

**Published:** 2022-08-09

**Authors:** Hongqin Yang, Yanjun Ma, Hongjie Zhang, Junyi Ma

**Affiliations:** College of Life Sciences, Northwest Normal University, Lanzhou 730070, China

**Keywords:** drug–protein interaction, human serum albumin, GDC-0941, molecular docking, molecular dynamics, spectroscopy, atomic force microscopy

## Abstract

Pictilisib (GDC-0941) is a well-known dual inhibitor of class I PI3K and mTOR and is presently undergoing phase 2 clinical trials for cancer treatment. The present work investigated the dynamic behaviors and interaction mechanism between GDC-0941 and human serum albumin (HSA). Molecular docking and MD trajectory analyses revealed that GDC-0941 bound to HSA and that the binding site was positioned in subdomain IIA at Sudlow’s site I of HSA. The fluorescence intensity of HSA was strongly quenched by GDC-0941, and results showed that the HSA–GDC-0941 interaction was a static process caused by ground-state complex formation. The association constant of the HSA–GDC-0941 complex was approximately 10^5^ M^−1^, reflecting moderate affinity. Thermodynamic analysis conclusions were identical with MD simulation results, which revealed that van der Waals interactions were the vital forces involved in the binding process. CD, synchronous, and 3D fluorescence spectroscopic results revealed that GDC-0941 induced the structural change in HSA. Moreover, the conformational change of HSA affected its molecular sizes, as evidenced by AFM. This work provides a useful research strategy for exploring the interaction of GDC-0941 with HSA, thus helping in the understanding of the transport and delivery of dual inhibitors in the blood circulation system.

## 1. Introduction

The physiological process of absorption, distribution, metabolism, and excretion of drugs in the body can directly affect the concentration of drugs in the action site and the duration of treatment, thus determining the occurrence, development, and elimination of pharmacological and toxicological effects of drugs. Many of the drug molecules are absorbed into the bloodstream, which carries the drug throughout the body where it can produce a therapeutic effect. Drugs are transported in the circulation either in a free form, dissolved in the aqueous phase of plasma, or complexed with plasma proteins in varying degrees [1,2]. This drug–protein binding is reversible in nature, and an equilibrium between the bound and free drug molecules exists [3,4]. Serum albumin is the most abundant circulatory protein in blood plasma and plays an important role in the transport and deportation of different types of endogenous and exogenous compounds to target cells [5,6]. Human serum albumin (HSA) is a principal extracellular protein with a concentration of 40 mg·mL^−1^ or 0.6 mM in blood plasma and contains 50–60% of total plasma protein in humans [7,8]. Research showed that the distribution, free concentration, and metabolism of drugs can be considerably altered as a result of their binding to HSA [9,10,11]. If the affinity of the drug binding to HSA in the plasma is strong, the drug will be transported to the target tissue and cannot be dissociated from HSA. Binding to HSA due to the weak affinity of drugs can lead to short active time or poor distribution in body. Initially, some researchers contended that drugs bind to HSA in a nonspecific behavior, but now, classes of binding domains appear to be limited. The most flexible and clear binding domains of HSA are located in the hydrophobic subdomains IIA and IIIA and are designated as sites I and II, respectively [12,13]. Therefore, the knowledge of the HSA–drug interaction is an important part of the overall drug development process and provides suitable information to determine ways in which the drug can be transferred.

The phosphatidylinositol 3-kinase/protein kinase B/mammalian target of rapamycin (PI3K/AKT/mTOR) pathway is the most frequently and aberrantly activated pathway in many types of cancers and plays a key role in cancer cell growth, survival, angiogenesis, and metastasis [14,15]. The PI3K/AKT/mTOR pathway has become an attractive therapeutic target for inhibiting the development and metastasis of tumors, resulting in the development and ongoing clinical trials of several PI3K/AKT/mTOR inhibitors [16,17,18]. Some inhibitors have begun to be used in clinical practice and have obtained varying degrees of success. However, some inhibitors have limited clinical potential because of high cytotoxicity, such as LY294002 [18]. Pictilisib (GDC-0941, C23H27N7O3S2, Figure 1) is a potent and selective dual inhibitor of class I PI3K and mTOR and shows favorable pharmacokinetic and toxicological properties when orally administered as a single agent or in combination with chemotherapy agents [17]. Pictilisib is currently undergoing phase II clinical trials for breast cancer [19], ovarian cancer [20], melanoma [21], and multiple myeloma treatment [22]. Clinical studies showed that GDC-0941 has a rapid absorption following oral administration and a dose-proportional pharmacokinetic profile [23]. Given the potential clinical applicative value of GDC-0941, research on its analogs (i.e., GDC-0084, GDC-0980, and GNE-317) has attracted attention [24,25,26]. Salphati et al. [27] found a weak interaction between GDC-0084 and plasma protein. Similarly, the preclinical assessment of absorption and disposition demonstrates that the binding of plasma protein to GDC-0980 is low, with the unbound fraction ranging from 29% to 52% across species [28]. Salphati et al. [29] previously reported that the binding of GNE-317 to plasma proteins is moderate, with a free fraction of 14.9% in mouse plasma. The majority of the recently published studies on these inhibitors are primarily aimed at preclinical metabolism, pharmacokinetics, and efficacious dose. As their representative, GDC-0941 has gained remarkable attention in cancer research over the past 10 years because of its 100-fold potency compared to other classes of dual PI3K/mTOR inhibitors [30]. Currently, no investigation seems to have been conducted on the interaction between GDC-0941 and HSA even though it may provide new insights into the plasma protein-binding characteristics in these classes of dual inhibitors.

The present work is a comprehensive in vitro study on the interaction of GDC-0941 with HSA by using various modern spectrum analytical technologies, molecular docking, and dynamic simulations, which thoroughly extend our knowledge of the interaction mechanisms involved in the dual PI3K/mTOR inhibitor–HSA binding process. Molecular docking and dynamic simulations were first used to evaluate the dynamic behaviors and stability of GDC-0941 binding to HSA and ascertain whether any particular binding property results in the interaction with a specific binding site on HSA. Steady-state and time-resolved fluorescence spectroscopy were applied to identify binding information, including quenching mechanisms, binding constants, number of binding sites, and possible binding forces. Warfarin sodium (WF-Na) and ibuprofen (IB) were used as site probes for locating the binding sites of the interaction of GDC-0941 and HSA. The effects of GDC-0941 on the microenvironment of specific amino acid residues and the conformation of HSA were examined through circular dichroism (CD), synchronous fluorescence, and 3D fluorescence spectroscopic methods. Moreover, the morphology of the protein molecule surface upon binding of GDC-0941 was studied by atomic force microscopy (AFM).

## 2. Results and Discussion

### 2.1. Molecular Docking Analysis

Molecular docking is proved to be an effective method for predicting the molecular recognition mode and determining the binding sites of the ligand molecule onto HSA [8,31]. As early as the 1970s, Sudlow et al. [12,13] provided authoritative findings on the binding sites of HSA and showed that some drug molecules with specific structural features are required for binding to Sudlow I or II. Crystallographic studies revealed that HSA is composed of three homologous α-helical domains (I–III), which have subdomains A and B [5,8]. Sudlow I and II are located in the hydrophobic subdomains IIA and IIIA and called sites I and II, respectively. AutoDock Vina was applied to visualize the interaction and predict the possible binding sites of GDC-0941 on HSA. The optimal binding conformation between GDC-0941 and HSA with the lowest binding energy of −8.1 kcal/mol was selected for discussion. Figure 2 shows that the whole GDC-0941 molecule entered the hydrophobic pocket (between subdomains IIA and IIIA) of HSA. Interestingly, the methylsulfonyl piperazine moiety of GDC-0941 bound into the subdomain IIIA at site II of protein, whereas the remaining moiety entered the subdomain IIA at site I. The GDC-0941 molecule in the binding site was surrounded by hydrophobic (i.e., LEU-198, PHE-211, TRP-214, ALA-215, and LEU-481), hydrophilic (i.e., GLN-196, CYS-200, and SER-202), and charged (i.e., LYS-199, ARG-218, and ASP-451) residues. Results suggested that the binding reaction was predominantly governed by hydrophobic forces and electrostatic interactions.

### 2.2. MD Trajectory Analysis

Based on the docking results, the HSA–GDC-0941 complex with the lowest binding energy was chosen for MD simulation. MD will be helpful in further examining the stability and dynamic behavior of the HSA–GDC-0941 complex and evaluating the reliability of docking results. Trajectories were recorded and analyzed during the whole simulation to obtain the RMSD, RMSF, Rg, binding energy, and number of hydrogen bonds. Given that RMSD can reflect the offset distance of all atoms in the conformation compared with the atoms in their initial position, it has been used extensively for assessing the stability of the simulation system. The RMSD values of the backbone atoms of protein and the heavy atoms of ligand in the HSA–GDC-0941 complex system are shown in Figure 3A. Results showed that the RMSD value displayed an apparent disturbance in the first 35 ns for the heavy atoms of GDC-0941 and subsequently reached a plateau. The RMSD value steadily increased for the backbone atoms of protein in the HSA–GDC-0941 complex system, and the increasing trend continued until approximately 75 ns. These findings suggested that the complex system reached equilibrium with 75–95 ns of MD simulation (Figure 3B) and that the stable combination could not be separated with the movement of molecules.

RMSF values of the HSA–GDC-0941 complex system were applied for analysis of the protein mobility based on the 100 ns trajectory time. As shown in Figure 3C, evident differences between the contours of the atomic fluctuations of the protein subdomain IIA and the other five subdomains were observed. Relatively, the RMSF values of the HSA–GDC-0941 complex in subdomain IIA were lower with little fluctuation, which indicated the increased conformational rigidity and stability of the HSA upon binding of GDC-0941 [32]. From a structural model point of view, subdomain IIA at site I was favorable for GDC-0941 binding to HSA. As illustrated in Figure 3D, the Rg values of the HSA–GDC-0941 complex system gradually achieved equilibrium after 65 ns simulation, which demonstrated that the dimension of protein increased due to the stable binding of GDC-0941 [31].

Total interaction forces were further decomposed to decipher the different energy components driving the binding of GDC-0941 (Appendix A). Based on the 75–95 ns target trajectories, 20,000 snapshots were saved, of which 200 snapshots were sampled and further used to calculate ∆*G*_bind_, Δ*E*_vdw_, Δ*E*_ele_, Δ*G*_GB_, and Δ*G*_SA_. Three decomposed energy components, namely Δ*E*_vdw_ (−58.4997 kcal/mol), Δ*E*_ele_ (−13.6927 kcal/mol), and Δ*G*_SA_ (−6.7683 kcal/mol), were beneficial to the binding of GDC-0941 to HSA, further demonstrating that the dominant forces were van der Waals and electrostatic interactions. Moreover, a decomposition of the calculated ∆*G*_bind_ value into contributions from each residue in the binding process. Table 1 shows that the last conformation of the binding of GDC-0941 to HSA via MD simulation at 95 ns was surrounded by residues SER-192, LYS-195, GLN-196, LEU-198, LYS-199, PHE-206, TRP-214, ALA-215, ALA-217, ARG-218, LEU-219, GLN-221, ARG-222, VAL-235, LEU-238, THR-239, VAL-241, HIS-242, CYS-245, and CYS-246. These residues were the key determinants of specific binding between GDC-0941 and HSA. Analysis of the frequencies of individual residue types involved in GDC-0941 binding shows that LYS, LEU, and ARG account for 16%, 19%, and 22% of the residues with the absolute contribution. It follows then that the impacts of LYS, LEU, and ARG are more important than other amino acids in the binding site of GDC-0941. The optimal bound conformations of GDC-0941 before and after MD simulation are displayed in Figure 4A. According to Figure 4A, the changes in the conformation of GDC-0941 were apparent, where the methylsulfonyl piperazine moiety of GDC-0941 was moved from subdomain IIIA to subdomain IIA in the final stable conformation. Compared with the results obtained from molecular docking, the hydrophobic interactions between GDC-0941 and five amino acid residues, i.e., Leu-198, Phe-211, Trp-214, Ala-215, and Leu-481, fractured and were rebuilt by the four other amino acid residues, i.e., Lys195, Lys199, Phe206, and Trp214 (Figure 4B). Other forces, such as π–cation interactions between the pyrimidine moiety of GDC-0941 and the LYS-199 residue, also maintained binding. Detailed information on interaction forces is presented in Table 2.

### 2.3. Quenching Mechanism Studies

Steady-state fluorescence spectroscopy, a simple and convenient method, can judge the occurrence of ligand–protein interactions and further provide vital information, such as quenching mechanism, binding constant, number of binding sites, and thermodynamic data [9,10]. The effect of GDC-0941 on the fluorescence intensity of HSA at 298 K is shown in Figure 5A. When the amount of GDC-0941 added to the HSA solution increased, the fluorescence emission peak at 337 nm had an evident redshift (~12 nm), with a gradually decreasing intensity. Results showed that GDC-0941 could interact with HSA and quench its intrinsic fluorescence due to their complexation. The redshift indicated that the microenvironment of the Trp-214 residue of HSA was affected by the presence of GDC-0941 [33]. HSA contained only one Trp in its structure located in the IIA subdomain. Therefore, findings were in good agreement with that obtained from MD simulation, and subdomain IIA was proposed to be the main binding pocket for GDC-0941 in HSA.

Fluorescence quenching could be broken down into static or dynamic quenching or both simultaneously, and the quenching mechanism could be distinguished by temperature dependence. Quenching-related data were analyzed using the classical Stern–Volmer equation, as follows [31,32]:(1)$$\frac{F_{0}}{F}=1+K_{q}\tau_{0}\left[Q\right]=1+K_{\mathrm{sv}}\left[Q\right]$$
where *F*_0_ and *F* are the fluorescence intensities of protein before and after quencher addition, respectively. *K*_q_ is the quenching rate coefficient (maximum value is 2.0 × 10^10^ L mol^−1^ s^−1^ for dynamic quenching) [31]. *τ*_0_ is the average fluorescent lifetime of protein without quencher (approximately 1.0 × 10^−8^ s) [31]. [*Q*] is the concentration of quencher. *K*_sv_ is the Stern–Volmer quenching constant. The Stern–Volmer curves at three different temperatures are shown in Figure 5B, and the corresponding curve slopes (*K*_sv_) are listed in Table 3. *K*_sv_ decreased accordingly with increasing temperature, which suggested that the overall quenching was dominated by a static quenching mode, forming an HSA–GDC-0941 complex. The *K*_q_ values in Table 3 (7.85, 7.31, and 6.18 × 10^12^ L mol^−1^ s^−1^) were calculated by *K*_sv_/*τ*_0_ and vastly exceeded the maximum scatter collision quenching constant (2.0 × 10^10^ L mol^−1^ s^−1^). Thus, the possibility of a dynamic quenching mode could be excluded.

Time-resolved fluorescence spectroscopy is the most effective way to distinguish the actual quenching mechanism between small molecules and biomacromolecules. The fluorescence lifetime does not rely on the quencher concentration in static quenching due to the formation of ligand–protein complex but decreases with increasing quencher concentration in dynamic quenching [10]. The fluorescence lifetimes of free HSA and the protein in the presence of GDC-0941 were determined to provide further evidence on the static quenching mechanism between GDC-0941 and HSA. All time-resolved fluorescence data were analyzed by exponential iterative fittings. The performance of fittings was estimated using chi-square (χ^2^) values; that is, χ^2^ < 1.3 was acceptable. The time-resolved fluorescence decay curves of HSA in the absence and presence of GDC-0941 are rendered in Figure 6. After adding a certain amount of GDC-0941, the fluorescence lifetime curve almost coincided with that of HSA alone. The average fluorescence lifetime (*τ*_avg_) was calculated on the basis of the respective decay times (*τ*_i_) and the relative amplitudes (*α*_i_) by using the following equation [10]:(2)$$\tau_{avg}=\tau_{1}\alpha_{1}+\tau_{2}\alpha_{2}+\tau_{3}\alpha_{3}$$

The fitted results are summarized in Appendix A. χ^2^ values were all quite close to 1, which indicated a satisfactory quality of exponential fittings. HSA showed tri-exponential fluorescence decay in PBS solution with the lifetime value of 5.4907 ns, and binding with GDC-0941 had almost no effect on *τ*_avg_ values. The disturbance of 1.4% was predominantly due to the increased GDC-0941 concentration. Results illustrated that the ground-state complex was formed between HSA and GDC-0941, and the quenching mechanism was static quenching. This result agreed well with those found in the above steady-state fluorescence measurements.

### 2.4. Binding Strength and Binding Force Studies

For the static quenching process, the association constant (*K*_a_) between GDC-0941 and HSA was calculated using the modified Stern–Volmer equation [31,32]:(3)$$\log\frac{F_{0}-F}{F}=n\log\left[Q\right]+\log K_{a}$$
where *n* is the number of binding sites. The log [(*F*_0_ − *F*)/*F*] versus log [*Q*] for the HSA–GDC-0941 complex system is plotted in Figure 5C, and the corresponding binding parameters at three different temperatures are listed in Table 3. The linearity of curves indicated that GDC-0941 could bind to only one site on HSA. The *K*_a_ values in the order of 10^5^ M^−1^ suggested a moderate binding strength between GDC-0941 and HSA, implicating that GDC-0941 could be stored and transported by the HSA in the body, which might be an important reason for GDC-0941′s good bioavailability after oral absorption. In addition, the *K*_a_ values of the HSA–GDC-0941 complex system presented a decreasing trend with increasing temperature due to the decomposition of the complex at high temperature. The moderate affinity was also observed in other published reports [9,10,33,34] and similarly interpreted in our previous study [31].

Four intermolecular interaction forces, including hydrophobic interactions, van der Waals forces, hydrogen bonding, and electrostatic interactions, play a key role in stabilizing the drug–protein complex. These interaction forces could be explained using the sign and magnitude of different thermodynamic parameters, including enthalpy (Δ*H*), entropy (Δ*S*), and free energy (Δ*G*) changes. According to the theory of Ross and Subramanian [35], negative Δ*H* and Δ*S* values indicated that van der Waals and hydrogen bond interactions played leading roles, whereas positive Δ*H* and Δ*S* values indicated that binding occurred through hydrophobic interactions. Furthermore, the positive Δ*S* and negative or small positive Δ*H* suggested that electrostatic interactions acted as main forces. The values of these parameters could be obtained by the Van ’t Hoff equation (Equation (4)) and Gibbs function (Equation (5)) as follows:(4)$$\ln K_{a}=-\frac{\Delta H}{RT}+\frac{\Delta S}{R},$$
(5)ΔG=ΔH−TΔS,
where *R* is the gas constant. The values of Δ*H* and Δ*S* could be attained from the slope and intercept of the Van ’t Hoff plot (Figure 5D) and calculated by Equation (4). Furthermore, Δ*G* was calculated using Equation (5). These values are listed in Table 3. The negative values of Δ*G* indicated a spontaneous binding reaction in the HSA–GDC-0941 complex. The observed negative values of Δ*S* and Δ*H* (−291.38 J mol^−1^ K^−1^ and −118.75 kJ mol^−1^, respectively) suggested that van der Waals and hydrogen bond interactions were the main binding forces to stabilize the complex and that the binding of GDC-0941 to HSA was an exothermic process. The binding rate decreased evidently with increasing temperature and was consistent with the result mentioned above that the binding constant *K*_a_ decreased with increasing temperature due to exothermic reaction. According to the results of the fluorescence assay and MD simulation, van der Waals forces were the vital forces involved in this binding interaction. However, other forces, such as electrostatic interactions and hydrophobic interactions, cannot be excluded.

### 2.5. Site Marker Displacement Studies

To further confirm the reliability of MD simulation results, we used two specific site markers, i.e., WF-Na (site I marker) and IB (site II marker), for the competitive binding between GDC-0941 and markers. Sudlow et al. [13] found that drugs bound to HSA induce different conformational changes in the albumin at sites I and II, which are detected by changes in the fluorescence and/or strength of binding of probes specific for the two sites. Therefore, a solution with fixed amounts of GDC-0941 and HSA was titrated in this research by successively adding WF-Na or IB. The percentage of displacement (*I*) by the corresponding site marker could be calculated as follows [13]:*I* (%) = *F*_2_/*F*_1_ × 100, (6)
where *F*_2_ and *F*_1_ are the fluorescence intensities of HSA–GDC-0941 without and with a site marker, respectively. Figure 7 illustrates the fluorescence intensities. *I* decreased remarkably with increasing WF-Na concentration, whereas a negligible effect was observed with IB addition. These results suggested that GDC-0941 and WF-Na had the same binding site in HSA. Thus, WF-Na competed with GDC-0941. Thus, GDC-0941 was bound to site I in subdomain IIA of HSA, and this finding was consistent with the abovementioned MD simulation.

### 2.6. Effect of GDC-0941 on the Conformation HSA

#### 2.6.1. CD Spectroscopy

CD spectroscopy is a quantitative technique for the rapid detection of the protein’s secondary structure and the conformation changes caused by ligand addition [7]. Numerous studies confirmed that the intermolecular or intramolecular forces that maintain the secondary or tertiary structures of protein may be affected by ligand binding [7,31]. As shown in Figure 8A, CD spectra were recorded for HSA with or without GDC-0941. The CD spectra of free HSA exhibited two negative absorption bands at 208 and 222 nm, representing the typical α-helical structure observed due to the *π*–*π** and *n*–*π** electronic transfers of the α-helix structure in the protein [36,37]. Figure 8A shows that the HSA band intensity at 222 nm increased in the presence of GDC-0941. However, no significant change in the peak positions and shapes of HSA were observed. This finding suggested that the α-helix structure was still predominant in HSA conformation after binding with GDC-0941. The observed CD values at 222 nm were sequentially analyzed using Equations (12) and (13) to quantify the contents of the α-helix structure of HSA. Free HSA had 53.94% α-helix structure, which was close to the reported values of 54.33% [36] and 54.42% [37]. The α-helix content of HSA with the addition of 3 μM GDC-0941 decreased from 51.98%. CD measurements indicated that the interaction of GDC-0941 with HSA caused a secondary change in the HSA and decreased the α-helix stability.

#### 2.6.2. Synchronous Fluorescence Spectroscopy

Synchronous fluorescence spectroscopy was carried out to interpret the change in the microenvironment of the amino acid residues. When Δλ between the excitation and emission wavelengths is stabilized at 15 and 60 nm, the maximum fluorescence emission peak in the synchronous fluorescence spectra reflects changes in the polarity around tyrosine (Tyr) and tryptophan (Trp) residues, respectively, of HSA [6]. Figure 8B shows that the fluorescence intensity of HSA evidently decreased at 15 and 60 nm on the successive addition of GDC-0941, suggesting the occurrence of emission quenching during the interaction between GDC-0941 and HSA. When Δλ was set to 60 nm, the synchronous fluorescence peak maximum slightly redshifted from 279 nm to 282 nm. However, no significant shift change was observed in the inset of Figure 8B when Δλ was set to 15 nm. These findings demonstrated that GDC-0941 had a higher effect on the microenvironment of the Trp residue in HSA compared with that for Tyr. Moreover, redshift effects suggested that GDC-0941 binding increased the polarity and decreased the hydrophobicity of microenvironments around the Trp residue. The redshift and decreasing intensity of the Trp residue were observed because only one Trp-214 residue was directly involved in the interaction of GDC-0941 with HSA.

#### 2.6.3. 3D Fluorescence Spectroscopy

Three-dimensional fluorescence spectroscopy is another powerful technique for studying perturbation on the conformation and microenvironment of protein upon ligand binding [38,39]. As shown in Figure 9, four peaks were observed in the 3D fluorescence spectra of free HSA and the HSA–GDC-0941 complex. Peaks A (λ_em_ = λ_ex_) and B (λ_em_ = 2λ_ex_) were ascribed to the first-order Rayleigh and second-order scattering peaks, respectively [39]. These peaks did not contain any relevant chemical information. Furthermore, peak I (λ_ex_ = 280 nm) predominantly provided the spectral characteristics of Tyr and Trp residues, and peak II (λ_ex_ = 230 nm) predominantly reflected the fluorescence characteristics of polypeptide backbone structures [39]. These peaks were correlated with the secondary protein structure. The peak position (λ_ex_/λ_em_), Stokes shift, and intensity representing characteristic information are listed in Appendix A. The fluorescence intensity of peak A had little change, whereas peak B increased with the addition of GDC-0941. This phenomenon could be attributed to the increased diameter of HSA caused by the binding with GDC-0941, which could enhance the Rayleigh scattering peak. After the addition of an excess of GDC-0941, the fluorescence intensity of peaks I and II showed an evident decrease by about 24.73% and 56.08%, respectively. Stokes shifts were observed in peaks I (redshift, 3 nm) and II (blueshift, 5 nm). Results of CD and synchronous fluorescence spectroscopy showed that the binding of GDC-0941 with HSA led to a change in polarity and hydrophobicity around the Trp residue and the disturbance of the polypeptide backbone structure.

### 2.7. Morphological Analysis of HSA

AFM is often used to observe the changes in the structure and morphology of a protein molecule’s surface upon binding of a drug owing to its spatial resolution [40]. AFM provides direct evidence of protein–drug interaction at molecular levels. In this study, 2D and 3D topography images were obtained to find out the size and morphology changes of HSA before and after GDC-0941 binding. From Figure 10A,C, once the free HSA and complex adsorbed on a silicon wafer, samples could be clearly detected, and the scan of the silicon slice revealed the accurate location. HSA formed some regular dots and rounded peaks in the absence of GDC-0941, as shown in Figure 10A,B. After averaging the width and height, the mean width of free HSA ranged from 154.11 nm to 138.03 nm, and the mean height ranged from 20.4 nm to 19.2 nm. After the addition of GDC-0941, apparent changes in the shape and size distribution points of HSA were observed. As shown in Figure 10C,D, high concentrations of GDC-0941 caused the flocculation or aggregation of the HSA molecule. These results clearly indicated that GDC-0941 could bind to HSA and transform the morphology of HSA from monomer to aggregated diploid. Previous studies found that protein aggregation affected the morphology and function of HSA [31,40].

## 3. Materials and Methods

### 3.1. Reagents

GDC-0941 (≥ 99%) was purchased from Selleck Chemicals LLC (Houston, TX, USA; http://www.selleckchem.com; accessed on 5 February 2022). Fatty-acid-free HSA was obtained from Sigma–Aldrich Corporation (Milwaukee, WI, USA). WF-Na and IB were purchased from J&K Scientific Ltd. (Beijing, China). All other analytical-grade reagents were used without purification. The solutions of HSA (20 μM), GDC-0941 (2 mM), phosphate-buffered saline (10 mM, pH 7.4), WF-Na, and IB (2 mM) were prepared before the experiment. Milli-Q water was used in all experiments. The stock solution of HSA was stored at 277 K.

### 3.2. Instruments and Operations

#### 3.2.1. Molecular Docking

The molecular docking study of GDC-0941–HSA in a simulated physiological environment was performed using AutoDock Vina. The 3D structure of GDC-0941 (PubChem CID: 17755052) was downloaded from the PubChem Compound database, and energy was minimized using the “Energy minimization” module in Chemdraw. The crystallographic structure of HSA (PDB: 1H9Z) was taken from the Research Collaboratory for Structural Bioinformatics Protein Data Bank (https://www.rscb.org; accessed on 14 March 2022). Before docking analysis, AutoDockTools was applied to optimize the crystal structures of HSA and GDC-0941 by removing all water molecules, adding Gasteiger–Hücker charges, assigning polar hydrogen atoms, and setting up rotatable bonds. HSA was rigid, whereas GDC-0941 was flexible during the docking process under optimum settings. During the docking process, the whole HSA was encompassed by setting the grid sizes along the X, Y, and Z axes to 120 Å, 120 Å, and 120 Å, respectively, with a grid box point spacing of 0.619 Å. Other parameters were set as default protocol settings. Docking results clustered around certain hot-spot conformations, and the lowest energy complex in each cluster was saved. The binding energies of all poses belonging to the current cluster were collected. The optimal conformation was selected on the basis of the highest binding energy to conduct the molecular dynamics (MD) simulation.

#### 3.2.2. MD Simulation

Once GDC-0941 had been docked into the active site, a 100 ns MD simulation was carried out with the Amber 16 software package by using GAFF and FF14SB force fields. Before MD simulation, the geometry structure of the HSA–GDC-0941 complex was optimized using the steepest descent and conjugate gradient methods after the system was heated to 300 K. During the MD simulation, the particle-mesh Ewald summation was used to calculate the long-range Coulomb interactions with a cutoff value of 8.0 Å. The SHAKE algorithm was applied to treat the stretching vibrations of associated hydrogen atoms. A step of 2 fs was used in the simulation, and data were collected every 10 ps. The conformational analysis was extracted from MD trajectories by the cpptraj module through means of the root mean square deviation (RMSD), root mean square fluctuation (RMSF), radius of gyration (Rg), and hydrogen bonds on critical trajectories. The binding free energy methods using the MM-GBSA module were applied to characterize the strength of the GDC-0941–HSA interaction, in line with past studies [41,42,43]. The binding free energy was computed on the basis of the following equations [42]:Δ*G*_bind_ = *G*_complex_ − (*G*_protein_ + *G*_ligand_),(7)
= Δ*E*_MM_ + Δ*G*_solvation_ − *T*Δ*S*,(8)
= Δ*E*_MM_ + Δ*G*_GB_ + Δ*G*_SA_ − *T*Δ*S*,(9)
= Δ*E*_vdw_ + Δ*E*_ele_ +Δ*G*_GB_ + Δ*G*_SA_ − *T*Δ*S*,(10)
where Δ*G*_bind_ is the free energy of binding (kcal/mol); Δ*E*_vdw_, Δ*E*_ele_, Δ*G*_GB_, Δ*G*_SA_, and *T*Δ*S* represent van der Waals forces, electrostatic interactions, polar solvent contribution to free energy of solvation, nonpolar solvent contribution to free energy of solvation, and entropy change upon ligand binding, respectively. The entropy caused by ligand binding (conformational entropy) was negligible due to the unusually high simulation cost. In addition, MM-GBSA analysis was also used to obtain a per residue energetic decomposition to define key binding determinants in the binding site.

#### 3.2.3. Fluorescence Measurements

The steady-state fluorescence and competitive site marker experiments were performed using an RF-5301PC Fluorescence Spectrometer (Shimadzu, Japan). The concentrations of HSA were fixed at 2 μM and added with different concentrations of GDC-0941 (0–9 μM). The wavelength range from 300 nm to 550 nm with excitation wavelength (*λ*_ex_) at 280 nm was selected. The widths of excitation and emission slits were 3 and 3 nm, respectively. Experiments were performed at 298, 304, and 310 K, using a thermostatic bath to maintain the temperatures. Considering the inner filter effect, all fluorescence intensities were correct in accordance with the following formula [9]:(11)$$Fc_{\;}=F_{m}e^{(A_{1}+A_{2})/2}$$
where *Fc* and *Fm* are the corrected and measured fluorescence intensities, respectively, and A_1_ and A_2_ are the absorbance values of the HSA–GDC-0941 system at excitation and emission wavelengths, respectively.

Site marker displacement experiments were designed to investigate the binding sites of GDC-0941 in the HSA–GDC-0941 solution (molar ratio = 1:1) at 298 K. The concentrations of WF-Na/IB ranging from 0 μM to 9 μM in the solution were obtained by gradually adding a 1.5 μM probe stock solution into the HSA–GDC-0941 solution. Fluorescence intensities were recorded under the same experimental conditions described earlier.

Synchronous and 3D fluorescence spectra were recorded using the Cary Eclipse Fluorescence Spectrophotometer (Varian, California, USA) at 298 K. The synchronous fluorescence spectra were obtained by considering wavelength intervals (Δ*λ* = *λ*_em_ − *λ*_ex_) at 15 and 60 nm. The HSA concentration was maintained at 2 μM in a quartz cell, and the concentrations of GDC-0941 were varied from 0 μM to 9 μM by successive additions. The 3D fluorescence spectra of HSA (2 μM) and HSA–GDC-0941 solutions (molar ratios = 1:1 and 1:3) were scanned through the use of an excitation wavelength ranging from 200 nm to 400 nm with 10 nm increments and monitoring the emission spectra between 200 and 500 nm.

#### 3.2.4. Time-Resolved Fluorescence Spectra

Time-resolved fluorescence spectra were obtained through the time-correlated single-photon counting technique with the Horiba Jobin Yvon FluoroMax-4 spectrofluorometer (Horiba Scientific, Paris, France) at room temperature. The spectra of free HSA (2 μM) and HSA–GDC-0941 (molar ratio = 1:1.5 and 1:3.0) were recorded using a quartz cell with a 1 cm path length at excitation and emission wavelengths of 280 and 345 nm, respectively.

#### 3.2.5. CD Spectroscopy

Far UV-CD measurements (200–260 nm) were carried out on a CD spectrometer (Model 400, AVIV, USA) equipped with the R3788 photomultiplier at 298 K. Measurements were conducted using a quartz cuvette with a path length of 0.1 cm. The molar ratio of HSA to GDC-0941 was fixed at 1:1.5, and the concentration of HSA was controlled at 2 μM. Each spectrum was the average of three successive scans. Results were expressed as mean residue ellipticity (*MRE*) in deg cm^2^ dmol^−1^. *MRE* was defined as follows [31]:(12)$$MRE_{222}=\frac{Intensity\;of\;\mathrm{CD}\;(\mathrm{mdeg})\;\mathrm{at}\;222\;\mathrm{nm}}{10\times C_{p}nl},\;$$
where *C*_p_ is the molar concentration of the protein, *n* is the number of amino acid residues (585 for HSA [28]), and l is the path length of the cuvette (0.1 cm). The α-helix contents of free HSA and HSA–GDC-0941 solution were evaluated by the following equation [31]:(13)$$\alpha -Helix(\%)=\frac{-MRE_{222}-2340}{30300}\times 100.$$

#### 3.2.6. AFM

AFM data were collected by the Multi-Mode Nanoscope V controller (Bruker, USA) with a 1.00 Hz scan rate at room temperature. HSA (3 μM) and HSA–GDC-0941 (1:3) solutions were dropped onto a fresh silicon wafer and incubated for 30 min at room temperature to allow the protein to adsorb onto the surface. Afterward, the silicon wafer was gently rinsed with triple-distilled water and air dried. Captured images (1 μm × 1 μm) were processed using NanoScope Analysis v.1.8.

## 4. Conclusions

In this paper, the binding characteristics, quenching mechanism, and conformation changes of GDC-0941 with HSA were delineated by theoretical models, AFM, and spectroscopic approaches. Theoretical models showed that GDC-0941 could combine with the hydrophobic cavity of the subdomain IIA of HSA (Sudlow’s site I) through weak noncovalent interactions, including van der Waals interactions, hydrophobic interactions, electrostatic interactions, and π–cation interactions. These findings were further confirmed by competitive experiments and thermodynamic parameter calculation results. The steady-state fluorescence and fluorescence lifetime values demonstrated that GDC-0941 formed a complex with HSA and quenched the fluorescence of HSA via a static mechanism. GDC-0941 showed good binding affinity to HSA with relatively high association constants (10^5^ M^−1^), suggesting the ability of albumin-mediated transport through the bloodstream. CD, synchronous fluorescence, and 3D fluorescence spectroscopy results showed that the microenvironments of the Trp residue and secondary structure of HSA changed and occurred in the HSA upon GDC-0941 insertion. AFM results revealed that the binding of GDC-0941 had certain influences on the molecular size and apparent morphology of HSA. Considering the momentous importance of HSA in drug transport, our results suggested that the interaction between GDC-0941 with HSA could affect the distribution of the dual inhibitor and cause side effects when taken into the body, resulting from conformational changes in the protein that could affect its function.

## Figures and Tables

**Figure 1 molecules-27-05071-f001:**
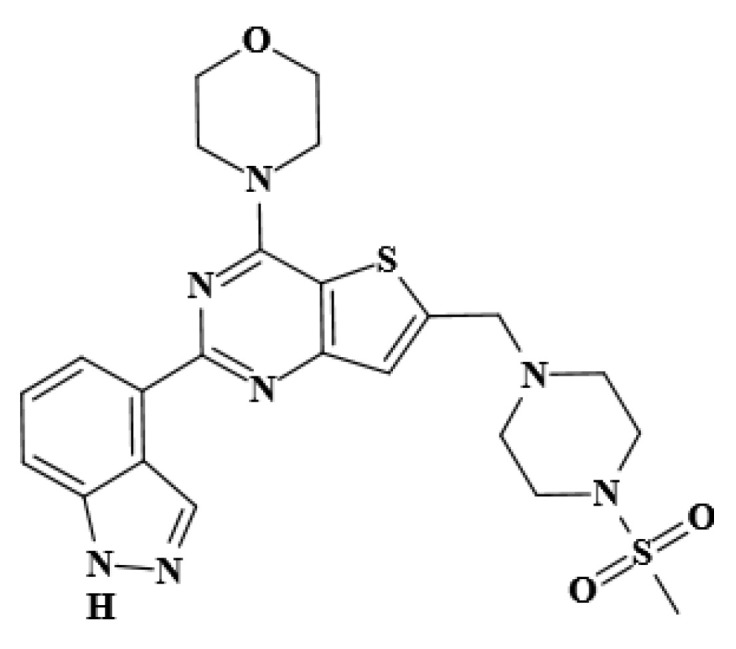
Chemical structure of GDC-0941.

**Figure 2 molecules-27-05071-f002:**
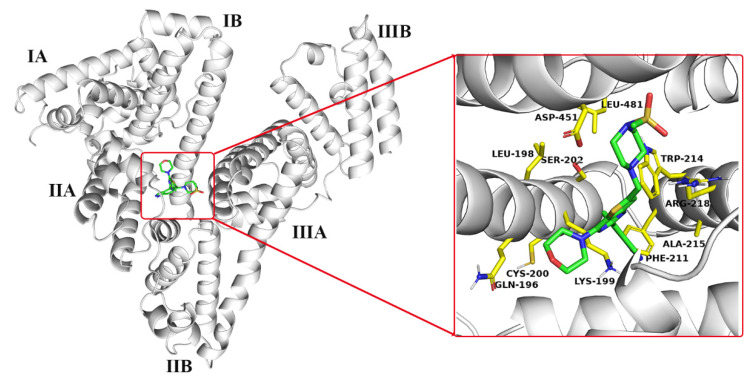
Molecular docking results of GDC-0941 binding to HSA. Left image shows the binding of GDC-0941 on HSA. Right image shows the view of the site of GDC-0941 binding to HSA. HSA is represented as a white cartoon; amino acid residues are represented as yellow sticks; GDC-0941 is represented as a green stick with nitrogen atoms in blue, oxygen atoms in red, and sulfur atoms in yellow.

**Figure 3 molecules-27-05071-f003:**
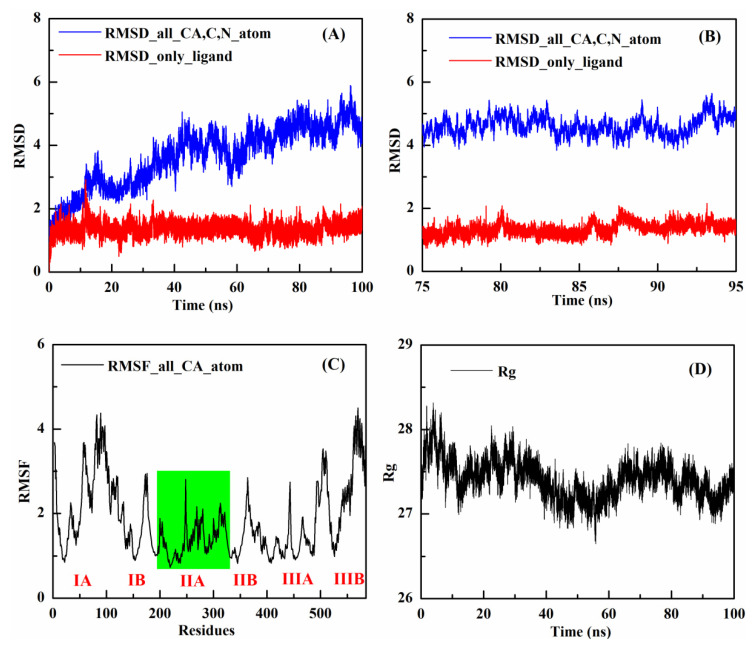
(**A**) The change curves of RMSDs of the backbone atoms of HSA (blue) and the heavy atoms of GDC-0941 (red) relative to the initial structure in the HSA–GDC-0941 complex system along the simulation time. (**B**) Partial enlarged view of the RMSD simulation time from 75 to 95 ns. (**C**) RMSF values of the HSA–GDC-0941 complex system plotted against residue numbers. (**D**) Time dependence of the Rg for the backbone atoms of HSA during the simulation in the presence of GDC-0941.

**Figure 4 molecules-27-05071-f004:**
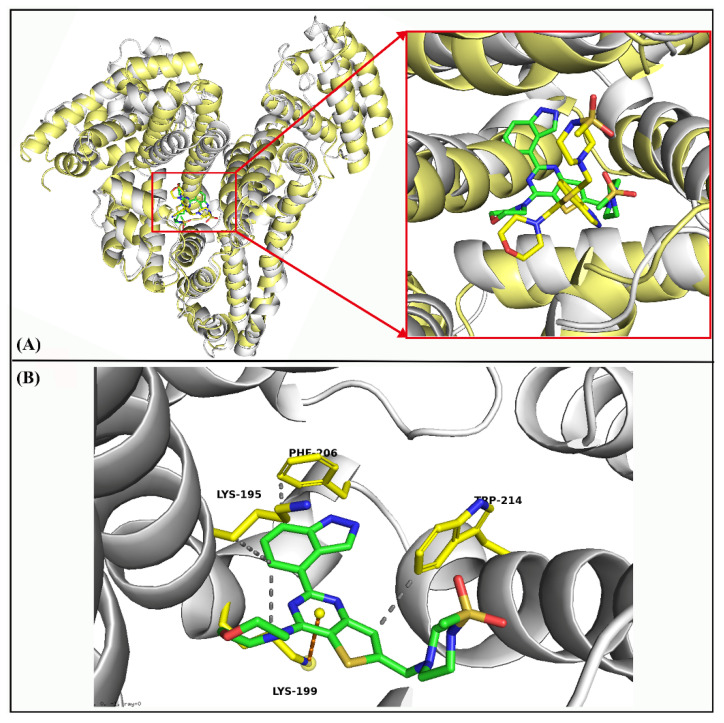
(**A**) Comparison of binding conformation between GDC-0941 and HSA before and after MD simulations of 95 ns. (**B**) The most stable conformation of the HSA–GDC-0941 complex system at 95 ns (hydrophobic interactions are depicted in gray and π–cation interactions in orange).

**Figure 5 molecules-27-05071-f005:**
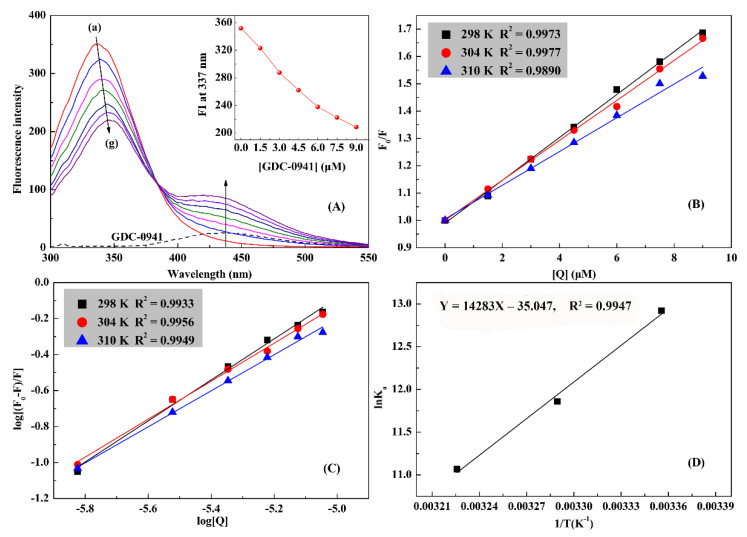
(**A**) Fluorescence emission spectra of HSA (2 μM) with increasing GDC-0941 concentrations (0–9 μM) at 298 K. The inset shows the fluorescence intensity (FI) of HSA at 337 in different concentrations of GDC-0941. (**B**) The curves of *F*_0_/*F* vs. [GDC-0941] at three different temperatures. (**C**) Plots of lg [(*F*_0_ – *F*)/*F*] against lg [GDC-0941] at different temperatures. (**D**) Van ’t Hoff plot of the HSA–GDC-0941 complex system.

**Figure 6 molecules-27-05071-f006:**
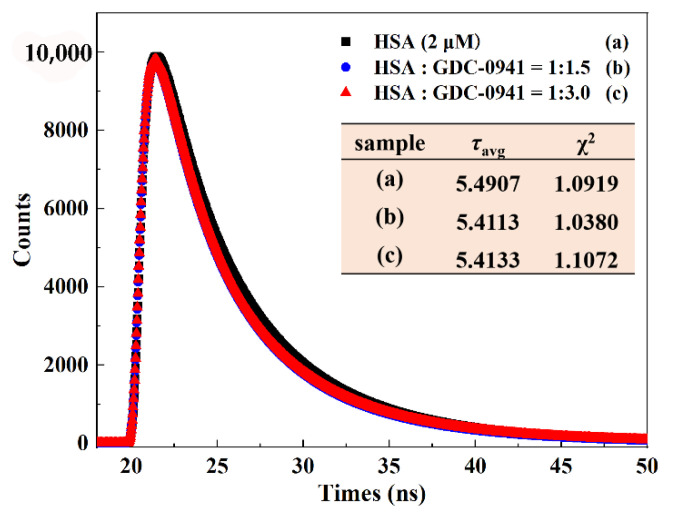
Time-resolved fluorescence lifetime of HSA (2 μM) in the presence of various concentrations of GDC-0941 (0, 3, 6 μM).

**Figure 7 molecules-27-05071-f007:**
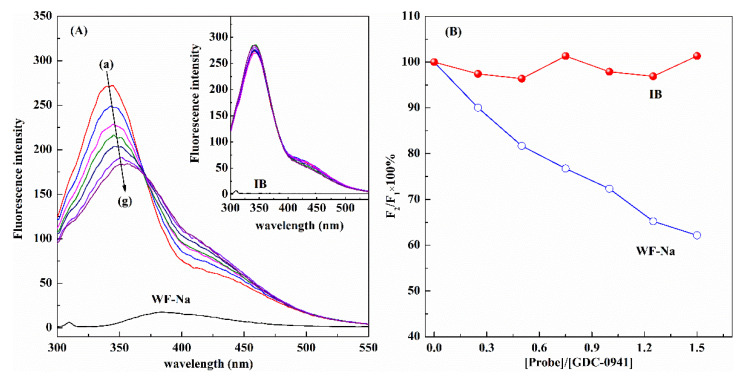
(**A**) Effect of WF-Na and IB on the fluorescence intensity of the HSA–GDC-0941 system (*C*_HSA_ = 2 μM, *C*_GDC-0941_ = 3 μM), λ_ex_ = 280 nm, pH = 7.4, *T* = 298 K. (**B**) The plot of values against the molar ratio of probe to GDC-0941.

**Figure 8 molecules-27-05071-f008:**
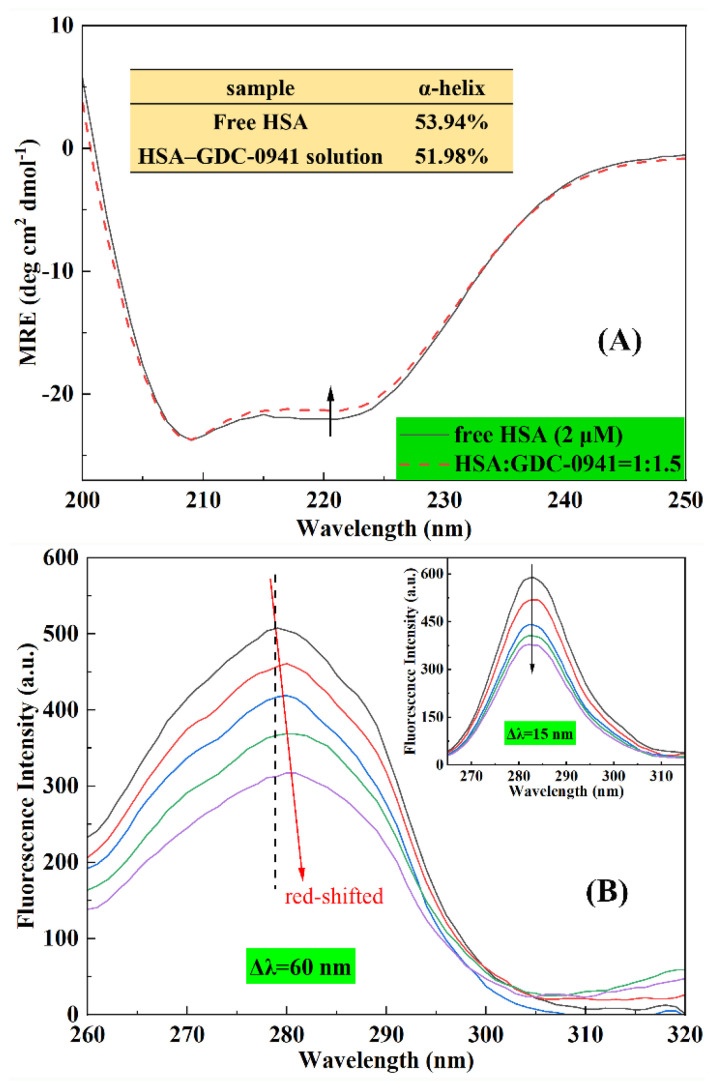
(**A**) CD spectra of HSA in the absence and presence of GDC-0941. Conditions: *C*_HSA_ = 2 μM; *C*_GDC-0941_ = 0, 3 μM; pH = 7.4; *T* = room temperature. (**B**) The synchronous fluorescence spectra of HSA in the absence and presence of GDC-0941 for Δλ = 60 nm. The inset represents Δλ = 15 nm. Conditions: *C*_HSA_ = 2 μM; *C*_GDC-0941_ = 0, 1.5, 3, 6, 9 μM; pH = 7.4; *T* = room temperature.

**Figure 9 molecules-27-05071-f009:**
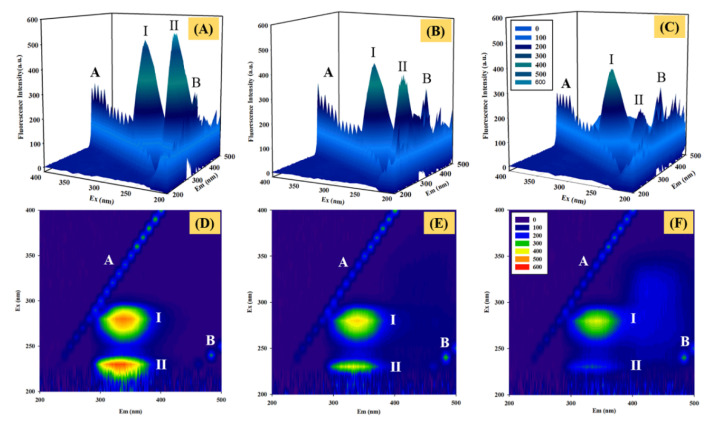
The 3D fluorescence spectra of free HSA (**A**) and HSA–GDC-0941 solution in the molar ratio of 1:1 (**B**) and 1:3 (**C**). Contour plots of 3D fluorescence spectra of free HSA (**D**) and HSA–GDC-0941 solution in the molar ratio of 1:1 (**E**) and 1:3 (**F**).

**Figure 10 molecules-27-05071-f010:**
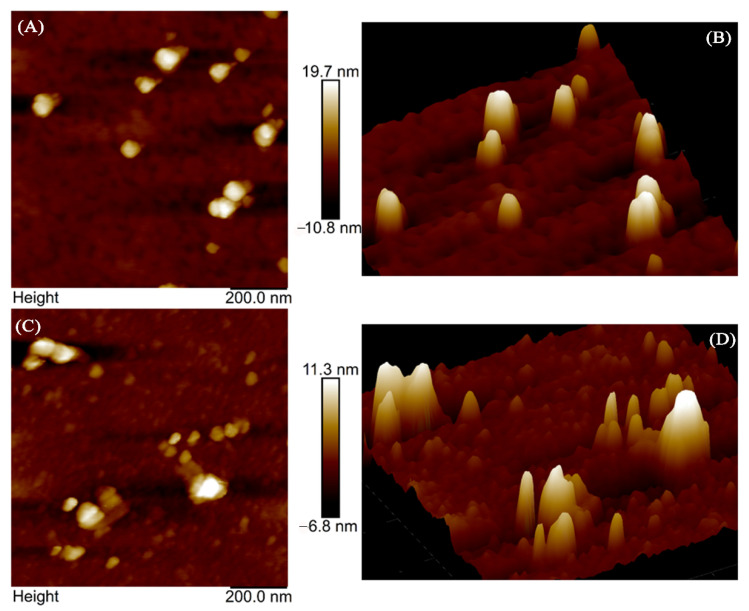
The 2D (**A**,**C**) and 3D (**B**,**D**) AFM images of HSA (3 μM) in the absence and presence of GDC-0941 (9 μM).

**Table 1 molecules-27-05071-t001:** MM-GBSA free energy of binding per residue decomposition. All values are in kcal/mol (# represents number).

HSA–GDC-0941 Complex					
Residue Name	Residue #	Contribution to ∆*G*_bind_	Residue Name	Residue #	Contribution to ∆*G*_bind_
SER	192	–0.37	LEU	219	–1.68
LYS	195	–1.94	GLN	221	–0.21
GLN	196	–1.76	ARG	222	–1.41
LEU	198	–0.49	VAL	235	–0.30
LYS	199	–1.09	LEU	238	–1.39
PHE	206	–0.95	THR	239	–0.38
TRP	214	–1.20	VAL	241	–0.29
ALA	215	–0.73	HIS	242	–1.06
ALA	217	–0.23	CYS	245	–0.35
ARG	218	–2.68	CYS	246	–0.31

**Table 2 molecules-27-05071-t002:** Detailed information of interaction forces in HSA–GDC-0941 complex at 95 ns.

Interaction Forces	Residue	Distance	Protein Charged?	Ligand Group/Atoms	Protein Atom
Hydrophobic Interactions	LYS195	3.94	Yes	C15–H	C
	LYS199	3.73	Yes	C15–H	C
	PHE206	3.59	No	C19	Ar–H
	TRP214	3.61	No	C8	Ar–H
π–Cation Interactions	LYS199	4.95	Yes	Aromatic (N3/5, C9/10/11/12)	N–H

**Table 3 molecules-27-05071-t003:** Association constants and thermodynamic parameters at different temperatures for the HSA–GDC-0941 interaction.

T (K)	*K*_sv_ (×10^4^ M^−1^)	*K*_q_ (×10^12^ L mol^−1^ s^−1^)	*n*	*K*_a_ (×10^5^ M^−1^)	Δ*G* (kJ mol^−1^)	Δ*H* (kJ mol^−1^)	Δ*S* (J mol^−1^ K^−1^)
298	7.85	7.85	1.14	4.09	−31.92	−118.75	−291.38
304	7.31	7.31	1.06	1.41	−30.17		
310	6.18	6.18	1.00	0.64	−28.42		

## Data Availability

Data sharing is not applicable to the paper; all supporting data are included within the main article.

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
