# Peer review of "PI3K/mTOR Dual Inhibitor Pictilisib Stably Binds to Site I of Human Serum Albumin as Observed by Computer Simulation, Multispectroscopic, and Microscopic Studies"

_molecules, 2022, doi:10.3390/molecules27165071_

Round 1
Reviewer 1 Report
The research is interesting and important in order to know about the importance of drugs distribution.
I suggest increase the references in the introduction about studies related with this research.
Author Response
The authors’ Answer: Special thanks to you for your good comments. According to your suggestion, the references “2. Agrawal R.; Thakur Y.; Tripathi M.; Siddiqi M.K.; Khan R.H.; Pande R.: Elucidating the binding propensity of naphthyl hydroxamic acid to human serum albumin (HSA): Multi-spectroscopic and molecular modeling approach. J. Mol. Struct., 2019, 1184, 1-11.”, “4. Amoorahim M.; Ashrafi-Kooshk M.R.; Esmaeili S.; Shahlaei M.; Moradi S.; Khodarahmi R., Physiological changes in the albumin-bound non-esterified free fatty acids critically influence heme/bilirubin binding properties of the protein: A comparative, in vitro, spectroscopic study using the endogenous biomolecules. Spectrochim. Acta A Mol. Biomol. Spectrosc., 2020, 235, 118298.”, “6. Li Z.; Zhao L.; Sun Q.; Gan N.; Zhang Q.; Yang J.; Yi B.; Liao X.; Zhu D.; Li H., Study on the interaction between 2,6-dihydroxybenzoic acid nicotine salt and human serum albumin by multi-spectroscopy and molecular dynamics simulation. Spectrochim. Acta A Mol. Biomol. Spectrosc., 2022, 270, 120868.” have been added. (Line 35, 36, 38, page 1).
In addition, the references “38. Maurya N.; Maurya J.K.; Singh U.K.; Dohare R.; Zafaryab M.; Moshahid Alam Rizvi M.; Kumari M.; Patel R., In Vitro Cytotoxicity and Interaction of Noscapine with Human Serum Albumin: Effect on Structure and Esterase Activity of HSA. Mol. Pharm., 2019, 16(3), 952-966.” (Line 358, page 11), “41. Rimac H, Tandarić T, Vianello R, Bojić M: Indomethacin Increases Quercetin Affinity for Human Serum Albumin: A Combined Experimental and Computational Study and Its Broader Implications. Int. J. Mol. Sci., 2020, 21(16):5740.”, “43. Samsonov SA, Pisabarro MT: Computational analysis of interactions in structurally available protein–glycosaminoglycan complexes. Glycobiology 2016, 26(8):850-861.” (Line 441, page 14) has been added.
Reviewer 2 Report
The article is well explained and the experimental and computational data reflect very well the reality of the interaction between the small ligand and the protein. Therefore, the article is in full condition for publication.
Author Response
The authors’ Answer: We are very grateful to reviewer #2 for his/her effort in reviewing our paper and his/her positive feedback.
Reviewer 3 Report
In this paper, the authors present computational and spectroscopic characterization of Pictilisib (GDC-0941) binding within the subdomain IIA (Sudlow's site I) of human serum albumin (HSA).
Kinetic and thermodynamic spectroscopic measurements were performed, which were completed with MD simulations of the drug bound to HSA. Initial structures were obtained by molecular docking.In addition, a conformational change of HSA was observed after binding Pictilisib (GDC-0941), which was confirmed by AFM microscopy.
This article is a positive example of how a combination of computational and experimental methods can yield very useful results, which can serve as a basis for a better understanding of the pharmacodynamics and pharmacokinetics of certain drugs.
I have only few comments:
1. Since I see that authors are using the MMGBSA module, I would suggest a decomposition of the Gibbs free energy of binding per residue (as in https://pubmed.ncbi.nlm.nih.gov/32785199/), which can provide even more direct confirmation of the importance of individual interactions.
2. In Fig. 4, please make cation - pi interactions more visible (sugestion: add pop up windov to existing picture to show interaction from side view and remove red arrow
Author Response
1. Since I see that authors are using the MMGBSA module, I would suggest a decomposition of the Gibbs free energy of binding per residue (as in https://pubmed.ncbi.nlm.nih.gov/32785199/), which can provide even more direct confirmation of the importance of individual interactions.
The authors’ Answer: Thank you for this valuable feedback. We have added the suggested content to the revised manuscript (Line 163-173, page 5). The reference “https://pubmed.ncbi.nlm.nih.gov/32785199/” has been referenced in Line 441, page 14.
2. In Fig. 4, please make cation - pi interactions more visible (suggestion: add pop up window to existing picture to show interaction from side view and remove red arrow.
The authors’ Answer: Thank you very much for your Suggestion. This has been corrected in the revised version of the manuscript, and a new Figure 4 has been included in the revised manuscript. (Please see the revised manuscript in page 6).